

**1**  **Effects of permafrost thaw on seasonal soil CO₂ efflux dynamics in a boreal**
**2**  **forest site**

**3**  Dragos A. Vas[1], Jaimie R. West[2], David Brodylo[1], Amanda J. Barker[1], William B. Baxter[1], and
**4**  Robyn A. Barbato[2]

**5**  [1] U.S. Army Engineer Research and Development Center-Cold Regions Research and Engineering
**6**  Laboratory, Ft. Wainwright, Alaska 99703, United States.
**7**  [2] U.S. Army Engineer Research and Development Center-Cold Regions Research and Engineering
**8**  Laboratory, Hanover, New Hampshire 03755, United State.

**9**  *Correspondence to* Dragos Vas (Dragos.A.Vas@usace.army.mil) ORCID: 0009-0005-2319-5079.

**10**  **Abstract.** Permafrost regions in subarctic and arctic areas harbor substantial carbon reserves, which are

**11**  becoming increasingly vulnerable to microbial decomposition as soils warm. As the seasonally thawed

**12**  active layer deepens and anthropogenic disturbances escalate, accurately predicting carbon fluxes from

**13**  thawed permafrost requires a comprehensive understanding of soil respiration dynamics. This study

**14**  aimed to investigate the impact of disturbance on soil respiration rates and identify the key environmental

**15**  and geochemical factors influencing these processes in a boreal forest ecosystem near Fairbanks, Alaska.

**16**  The disturbed site demonstrated an increase in mean annual soil temperatures, recorded at 0.60 ± 0.16°C,

**17**  along with a 14.4% rise in mean annual microbial activity, which peaked at 20% during the summer, in

**18**  contrast to the undisturbed site, which had a mean annual temperature of -0.37 ± 0.08°C. Furthermore,

**19**  bacterial and fungal community composition differed significantly between the two sites, suggesting a

**20**  potential mechanism underlying the variation in CO₂ efflux. Our research underscores the essential

**21**  importance of considering the rise in carbon emissions from anthropogenically disturbed soils in

**22**  permafrost areas, which are frequently neglected in assessments of the carbon cycle. This study

**23**  contributes to a deeper understanding of the complex interactions governing soil respiration in thawing

**24**  permafrost, ultimately informing more accurate predictions of carbon fluxes in these ecosystems.

**25**  **1. Introduction**

**26**  Soil respiration, the process by which carbon dioxide (CO₂) is released from the soil surface to

**27**  the atmosphere, is a critical component of the global carbon cycle. This process is influenced by the

**28**  microbial breakdown of organic material as well as the respiration of plant roots. Understanding soil

**29**  respiration dynamics is particularly crucial in boreal forests, as they encompass approximately 30% of the

**30**  global forest and play a vital role in global carbon sequestration (Bonan, 2008; Pan *et al*., 2011; Chi *et al*.,

**31**  2021). Recent studies indicate that increasing temperatures could lead to boreal forests transitioning from

**32**  functioning as carbon sinks to becoming carbon sources (Bond-Lamberty *et al*., 2018; Marty *et al*., 2019,



Harel *et al.*, 2023). In boreal forests, soil respiration is estimated to contribute up to 68% of the total
ecosystem respiration (Parker *et al.*, 2020; Watts *et al.*, 2021) and is significantly impacted by changes in
soil temperature, soil moisture, the microbial community, and the type of vegetation present (Grace, 2004;
Fekete *et al.*, 2014; Rodtassana *et al.*, 2021). Soil respiration in permafrost regions is also influenced by
the deepening of the active layer (seasonal surface soil thaw) that may occur due to permafrost
degradation exacerbated by warmer soil and air temperatures (Turetsky *et al.*, 2020; Watts *et al.*, 2021).

Anthropogenic (e.g., trail development, firewood harvesting) and natural (e.g., wildfires,

flooding, drought) disturbances can significantly impact the terrain by altering the ground vegetation, as
well as soil temperature, structure, water content, organic matter content, and microbial communities,
thereby influencing the rates and patterns of soil respiration. Previous studies conducted in boreal forest
and spruce forest environments have shown that tree harvesting leads to a long-term increase in $CO_2$
effluxes due to a decrease in net radiation (Amiro 2001) and an increase in soil temperature (Gordon *et*
*al.*,1987; Lytle and Cronan 1998; Amiro 2001) and soil water content because of vegetation cover loss
(Halim *et al.*, 2024). Tree harvesting also has a profound effect on the availability of substrates, both in
terms of quantity and quality, which subsequently modifies the biomass and structure of the microbial
community (Chatterjee *et al.*, 2008).

In a carbon dynamics study conducted in three boreal fen peatlands of Ontario, Canada, Webster

*et al.* (2023) discovered that drought conditions can significantly elevate $CO_2$ efflux while simultaneously
reducing the $CH_4$ efflux. Additionally, they found that shallow flooding can lead to a reduction in $CO_2$
emissions and an increase in $CH_4$ emissions. A separate investigation revealed that the flooding
significantly influenced the dynamics of $CO_2$ and $CH_4$ in riparian forests subject to varying degrees of
flooding. $CO_2$ emissions tend to rise with increased flood frequency, averaging 1.6 times higher in flood-
affected riparian forests compared to those protected from flooding, which exhibited unexpectedly strong
$CH_4$ sink characteristics (Jacinthe, 2015). Similar to the effects of flooding and drought, wildfires in
boreal forests significantly influence soil carbon efflux. Following the fire, the efflux of $CO_2$ from the soil
experienced a decline (Amiro, 2001; Koster *et al.*, 2017 and 2018; Halim *et al.*, 2024); however, it
subsequently rose over the course of several decades, ultimately peaking approximately 40 to 45 years
later. (Halim *et al.*, 2024).

Research on changes to soil respiration in disturbed environments is crucial to enhance our

understanding of the resilience and vulnerability of boreal forests to disturbances. As global temperatures
rise, the frequency and intensity of disturbances in these regions are expected to increase, potentially
leading to significant changes in soil carbon fluxes. Further, insights gained from such studies can inform



forest management practices aimed at mitigating the impacts of disturbances and preserving the carbon
sequestration potential of boreal forests.

The aim of this study is to measure and compare the soil respiration rates in an undisturbed

subarctic boreal forest with those in a subarctic boreal forest that has been affected by historical activities
tied to mining such as trail development and firewood harvest. These disturbances took place in the early
1900s, coinciding with the construction of a drainage ditch and an access trail to support mining
operations. Currently, there is no active drainage at the research site, and the trail is seldom used. The
study seeks to reveal the fundamental mechanisms of soil respiration in these ecosystems by examining
various edaphic factors, including soil temperature, moisture content, soil organic matter (SOM), pH, and
the composition of microbial communities. A Random Forest Model (RFM) was utilized in conjunction
with regression analysis to analyze the time series data, which encompassed variables such as soil
respiration, temperature, moisture, and air temperature. Furthermore, R statistical ANOVA analysis was
performed to evaluate soil characteristics, particularly pH and SOM, as well as the composition of soil
microbial communities. We hypothesize that there is a considerable escalation in carbon emissions from
anthropologically disturbed soils as compared to undisturbed soils in permafrost areas and we propose
that these emissions must be incorporated into models of carbon fluxes within these ecosystems.
**2. Materials and Methods**
**2.1 Site description**

The study was conducted at two adjacent sites underlain by permafrost in a subarctic boreal forest

located at the U.S. Army Cold Regions Research and Engineering Laboratory (CRREL) Permafrost
Research Tunnel Facility in Fox, Alaska (64.9507 N -147.6200 W, 248 m a.s.l.). The region experiences a
continental climate, which is defined by an average annual air temperature of −2.4 °C, with average
temperatures in July reaching 16 °C and January temperatures averaging −21.9 °C; extreme temperatures
throughout the year can range from −51 °C to 38 °C (Jorgenson et al. 2020). The two sites were situated
approximately 10 m apart; the first consists of an undisturbed black spruce forest ecosystem; the second
consists of an anthropogenically disturbed area where trails were established, and firewood was harvested
as part of mining activities in the region in the 1920's.

The vegetation at the undisturbed site consists of small black spruce (*Picea mariana*) ranging

from densely distributed to tightly spaced. Understory canopy is dominated by marsh and bog Labrador
tea (*Rhododendron tomentosum*; groenlandicum). Forest floor cover is primarily mosses (feather mosses
and *Sphagnum spp.*) and small shrubs including lowbush cranberry (*Vaccinium vitis-idaea*). The disturbed



site is characterized by scattered birch (*Betula neoalaskana*) and white spruce (*Picea glauca*) cover. The
understory canopy is primarily dwarf shrubs including marsh Labrador tea and bog blueberry (*Vaccinium*
*uliginosum*). The ground surface cover is dominated by grasses (*Poaceae*) and sedges (*Cyperaceae*). Soil
material types were classified as mineral soil (<20% OM; Soil Survey Staff, 2022) with more organic-rich
fractions (10-18% OM) comprising a surface layer (topsoil) and lower organic content (<5% OM) in the
stratigraphically deeper subsoil. The topsoil textures ranged from loam to silt loam, reflecting a higher
proportion of sand particles in the topsoil relative to the silt loam subsoil (SI Table 1).

### 2.2 Data Collection

Total soil respiration (autotrophic and heterotrophic), temperature, volumetric water content
(VWC), air temperature, and barometric pressure were measured from 4 Nov 2022 to 9 Nov 2023 at eight
plots; four plots were located at the undisturbed site and the remaining four at the adjacent disturbed site.
In preparation for total soil efflux measurements, a soil collar made from thick-walled polyvinyl chloride
(PVC) pipe was inserted at each of the eight plots. The collars had an inside diameter of 21.3 cm and a
height of 11.4 cm and were inserted 2-3 cm into the soil through the soil vegetation cover. Soil
temperature and VWC sensors were also installed in both the topsoil and subsoil layers at all plots.
The varying temporal resolution soil parameter (30 min), weather (15 min), and soil efflux (30
min) data were averaged to hourly, daily, and seasonal means for statistical analyses. The seasons were
categorized as winter (November to March), spring (April to May), summer (June to August), and autumn
(September to October), based on the observed patterns of efflux seasonality at the research location. This
classification aligns with earlier studies on boreal forest efflux conducted by Wats et al. (2021) in Alaska
and Canada.

### 2.3 Total soil $CO_2$ efflux measurements

The total soil $CO_2$ efflux respiration, which encompasses the overall release of $CO_2$ from the soil
into the atmosphere (including both autotrophic and heterotrophic processes), was measured using a LI-
COR Soil gas flux system (LI-COR inc., Lincoln, Nebraska, USA) composed of 2 gas analyzers, LI-870
$CO_2/H_2O$ analyzer and LI-7810 $CH_4/CO_2/H_2O$ trace gas analyzer, a LI-8250 multiplexer, and eight 8200-
104 Opaque Long-Term Chambers. The two gas analyzers and multiplexer were housed in a custom-made
Styrofoam enclosure (Figure 1a) to protect them from precipitation and extreme winter temperatures. The
eight chambers were deployed over the soil covers installed at the study plots and were connected to the
multiplexer by a 15 m long tubing and cable assembly provided by the manufacturer. Minor adaptations
to the soil gas flux system were necessary to ensure data collection in the winter months; first the cable



and tubing assembly was wrapped in 1.3 cm thick tubular pipe insulation foam to prevent/minimize
clogging due to moisture freezing up inside the tubing, and second, Costco (Columbus, Indiana) folding
tables with a surface of 91.4 cm × 91.4 cm were positioned over the 8200-104 long-term chambers to
prevent snow accumulation on the chambers (Figure 1b). Therefore, all sample locations were equally
affected by the lack of snowfall and possible grazing by animals.  A detailed description of the soil gas
flux system operation and winter modifications can be found in Vas *et al*. (2023).

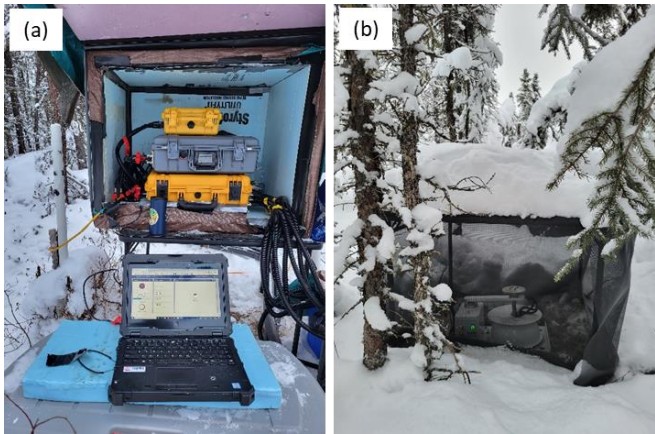


**134 Figure 1. Custom modifications for LI-COR Soil gas flux system in cold climates. Specially designed**
**135 enclosures to ensure optimal operating temperatures (a) (photo credits: Dragos A. Vas, 17**
**136 November 2023), insulated tubing for instruments to prevent clogging, and long-term covers for**
**137 chambers (b) (photo credits: Dragos A. Vas, 17 November 2023) to inhibit snow accumulation or**
**138 drifting on the chambers.**


**140 2.4 Soil and meteorological condition measurements**

Soil temperature, VWC, air temperature, and barometric pressure were measured using the
following Onset HOBO (Onset, Bourne, Massachusetts, USA) instrumentation: U30 USB Weather
Station, S-TMB-M002 12-Bit Temperature Smart Sensor, S-SMC-M005 EC5 Soil Moisture Smart
Sensor, S-THC-M002 Temperature/Relative Humidity Smart Sensor, and S-BPB-CM50 Smart
Barometric Pressure Sensor. Soil temperature and VWC were recorded in the topsoil and subsoil layers,
approximately 30 cm from each long-term chamber, at a depth of $18.5 \pm 1.19$ cm in the topsoil layer and
$34.5 \pm 1.47$ cm in the subsoil layer for the undisturbed site. At the disturbed site, the measurements were
taken at $13.5 \pm 2.53$ cm in the topsoil layer and $35.5 \pm 1.85$ cm in the subsoil layer. These depths were
measured from the top of the ground vegetation cover, which had a thickness of $16.5 \pm 1.19$ cm at the
undisturbed site and $9.5 \pm 0.87$ cm at the disturbed site.  VWC measurements were restricted to the



summer and autumn seasons due to the sensor's inability to measure below freezing temperatures. Air
temperature and barometric pressure were measured at 2 m above the substrate surface.

The depth of thaw in the active layer was assessed at ~ 10 cm from each of the eight chamber

plots during each site visit (n = 160) from 12 May to 3 Oct 2023. To determine this depth, a graduated
metal rod with a diameter of 1 cm (known as a frost probe) was inserted into the ground, at the same
location, until it met resistance, establishing the distance between the ground surface vegetation and the
top of the frozen soils (Shiklomanov *et al*. 2013).
**2.5 Soil collection and property analysis**

Soil samples were collected from both the topsoil and subsoil layers across all plots in the autumn

(September 2022), winter (February 2023), and summer (June 2023) seasons to analyze potential
variations in microbial community composition among seasons, disturbance regimes, and soil layers. The
samples were collected at identical depths during each season, which coincided with the depth at which
the soil temperature and moisture probes were positioned; these depths varied for each plot. The winter
samples were acquired using a gas-powered SIPRE (Snow, Ice, and Permafrost Research Establishment)
corer (Jon's Machine Shop, Fairbanks, Alaska, USA). Nitrile gloves were utilized to minimize any
potential contamination to the cores. Furthermore, the SIPRE corer and all associated tools were
thoroughly sanitized with 70% isopropyl alcohol, DNA away, and RNase away (Thermo Fisher Scientific
in Waltham, MA, USA). The cores were then subsampled into approximately 5 cm long cylinders using a
sanitized hammer and chisel and were carefully placed into sterile Nasco™ Whirl-pak bags (Thermo
Fisher Scientific, Waltham, MA, USA); further information on this sampling method can be found in
Barbato *et al*. (2022). Summer and autumn soil samples were gathered using a sanitized trowel with 70%
isopropyl alcohol, DNA away, and RNase away. The samples, approximately 5 cm thick, were placed in
sterile Nasco™ Whirl-pak bags and immediately placed in a cooler with frozen ice packs, then transferred
to a freezer upon arrival at the Cold Regions Research and Engineering Laboratory in Fairbanks, Alaska
(CRREL-AK). All collected soil samples were kept at a temperature of -25 °C until shipped to CRREL in
Hanover, New Hampshire (CRREL-NH), where they were stored at -20 °C until further processing.

Loss on ignition (LOI) was measured as a proxy for soil organic matter (SOM) content (Storer,

1984) on all soil samples. Here, LOI is the proportion of mass loss from oven-dried soil (dried at 105 °C
for 24 hours) following 2 hours at 360 °C in a muffle furnace. Soil total carbon and total nitrogen was
measured via combustion using a TruSpec C and N Analyzer (LECO, St. Joseph, MI, USA) at the
University of Wisconsin Soil and Forage Lab. Soil pH was measured from a 1:1 slurry of soil:CaCl$_2$
solution (0.01M) using a pH probe (Hanna Instruments, Woonsocket, RI, USA) and a SevenEasy S20 pH




meter (Mettler Toledo, Columbus, OH, USA). Soil pH was converted to H+ concentration prior to taking an average or statistical analysis. LOI total carbon, total nitrogen and soil pH was statistically analyzed using ANOVA in R.

**2.6 Soil microbial DNA extraction, gene sequencing, and data analysis**

Soil was partially defrosted and homogenized in the sample bag prior to subsampling 250 mg into bead beating tubes. Total genomic DNA was extracted using the DNeasy PowerSoil Pro Kit (Catalog No. 47014, Qiagen, Germantown, MD, USA), using a Precellys Evolution Touch homogenizer for the bead beating step (Catalog number P002511-PEVT0-A.0, Bertin Technologies, Montigny-le-Bretonneux, France). Automated DNA extraction was done with a QIAcube Connect (Catalog No. 9002864, Qiagen, Germantown, MD, USA) and each extraction run included a blank. Extracted DNA was held at -20 °C.

Library preparation and sequencing was completed at Argonne National Laboratory (Lemont, IL, USA), as follows. For bacterial analysis, the V4 region of the 16S rRNA gene was targeted for PCR amplification with region-specific primers (forward primer 515F and reverse primer 806R); and for fungal analysis, the ITS region was amplified using appropriate barcoded primers (Caporaso *et al*., 2011; Caporaso *et al*., 2012; Apprill *et al*., 2015; Parada *et al*., 2016; Smith *et al*., 2014; Walters *et al*., 2016). Each PCR reaction contained 1 µL template DNA, 12.5 µL AccuStart II PCR ToughMix (Quantabio, Beverly, MA, USA), 1 µL forward primer with Golay barcode (5 µM concentration), 1 µL reverse primer (5 µM concentration), and 9.5 µL DNA-free PCR water. PCR conditions were: 94 °C (3 minutes to denature the DNA); 35 cycles of 94 °C (45 s), 50 °C (60 s), and 72 °C (90 s); final extension at 72 °C (10 minutes). PCR product was quantified using Quant-iT PicoGreen (P7589, Invitrogen, Waltham, MA, USA). Equimolar amounts of amplicons were pooled, purified using AMPure XP Beads (A63881, Beckman Coulter, Brea, CA, USA), quantified (Qubit, Invitrogen), and diluted to 6.75 pM using a 10% PhiX spike. Paired-end 2 x 251 sequencing was done on a MiSeq (Illumina, San Diego, CA, USA).

Sequencing data were processed in R (R-Core-Team, 2018), using a dada2 v1.18.0 pipeline (Callahan et al., 2016), as in Baker et al. (2023), implemented using Snakemake v7.25.0 (Mölder et al., 2021). Taxonomy assignment was based on the SILVA 138.1 reference database for 16S sequences (Quast *et al*., 2013; Yilmaz et al., 2013) and the UNITE database (release 25.07.2023) for ITS sequences (Kõljalg *et al*., 2013; Nilsson *et al*., 2019). Chloroplasts and mitochondria were excluded from the dataset. A total of 2836 fungal ASVs and 10,608 bacterial ASVs were identified (excluding extraction blanks). Amplicon sequences are in the National Center for Biotechnology Information Sequence Read Archive (NCBI SRA), accession PRJNA1178745. Though our bacterial primers targeted both bacterial and archaeal 16S rRNA, we will refer simply to bacteria, which comprise 99.8% of total reads. Of archeal reads, 80%



represented the phylum *Crenarchaeota*. One sample was excluded from analysis because it was
mislabeled (2023_Feb, Chamber 3, Organic).
R (R-Core-Team, 2018), and *ggplot2* (Wickham, 2016) were used for data analysis and
visualization; the bioinformatic approach followed West et al. (2022). Community composition was
visualized using principal coordinates analysis (PCoA) of Bray-Curtis dissimilarities (Bray and Curtis,
1957) generated using *avgdist* from the R package *vegan* (Oksanen *et al*., 2024) using rarefaction (999
iterations) to a sampling depth of 16,300 for 16S and 6150 for ITS. A significant effect ($p < 0.05$) of
disturbance, sampling date, and interaction of these factors on community composition was tested within
each soil layer (topsoil and subsoil), using permutational multivariate analysis of variance
(PERMANOVA; *adonis2* from *vegan*) (Anderson, 2001). Richness was evaluated using weighted linear
regression (*betta* function in *breakaway* R package) (Willis *et al*., 2017), and a significant effect of
disturbance tested via ANOVA. Differential abundance (*differentialTest* in the *corncob* package) (Martin
*et al*., 2021) was then used to identify significant enrichment or depletion of individual taxa due to
disturbance, after excluding taxa with mean relative abundance < 0.00001.
**2.7 Random Forest modeling**
A regression-based Random Forest (RF) model developed in R was used to identify the relative
importance of the input variables to predict hourly and daily $CO_2$ concentrations. RF was chosen over
other algorithms due to its wide and successful application in determining variable importance
(Behnamian *et al*., 2017; Lei *et al*., 2024). In RF, the supervised non-linear algorithm can combine
predictions from hundreds or thousands of individual decision trees via bootstrap aggregation to generate
an ideal output (Schonlau and Zou, 2020). Compared with individual decision trees, this results in an
increase in generalization accuracy and a reduction in overfitting. A repeated k-fold cross-validation
technique was also employed. In this technique, data are randomly separated into $k$ subsets with $k$-1 used
to train the model and the remainder to test the model, which is then repeated a specified amount. We
selected a value of 10 for k and a value of 5 for repetition. Input variables were the same for each instance
except barometric pressure being dropped for daily $CO_2$ concentrations due to poor importance values.
Thaw depth was static in the dataset that the model used from 4 October through 16 May due to the
presence of a frozen surface layer preventing thaw depth probing. Organic soil VWC and mineral soil
VWC from 1 November – 31 March and 1 April – 31 May were omitted for machine learning due to the
inability of the probes to function properly in subzero temperatures.
**3. Results**



**3.1 Soil conditions**


Soil temperature fluctuated by a factor of three at the disturbed site and by a factor of 1.9 at the
undisturbed site over the course of the year. The undisturbed and disturbed sites exhibited contrasting
thermal regimes. At the undisturbed site, the mean annual soil temperatures were below freezing,
exhibiting -0.33 ± 0.1 °C for topsoil and -0.41 ± 0.07 °C for subsoil (Figure 1). In contrast, the disturbed
site experienced positive mean annual soil temperatures, with 0.72 ± 0.2 °C for topsoil and 0.48 ± 0.13 °C
for subsoil. Winter was the only season with warmer topsoil and subsoil temperatures at the undisturbed
site. For both the undisturbed and disturbed sites, the subsoil layer was cooler than the topsoil layer in
terms of mean annual temperature. Significantly warmer soil temperatures were observed at the disturbed
site during the summer (4.04 ± 0.19 °C; $p$ <0.001, ANOVA) and autumn (1.88 ± 0.23 °C; $p$ <0.001,
ANOVA) in comparison to the temperatures recorded at the undisturbed site during the same seasons.
Conversely, the mean summer temperature at the undisturbed site was 1.15 ± 0.08°C, while the mean
autumn temperature was 0.38 ± 0.07 °C (Figure 2). Soil temperatures in the shallower topsoil layer
exhibited greater variability throughout the year compared to the temperatures in the deeper subsoil layer
at both sites (Figure 2).

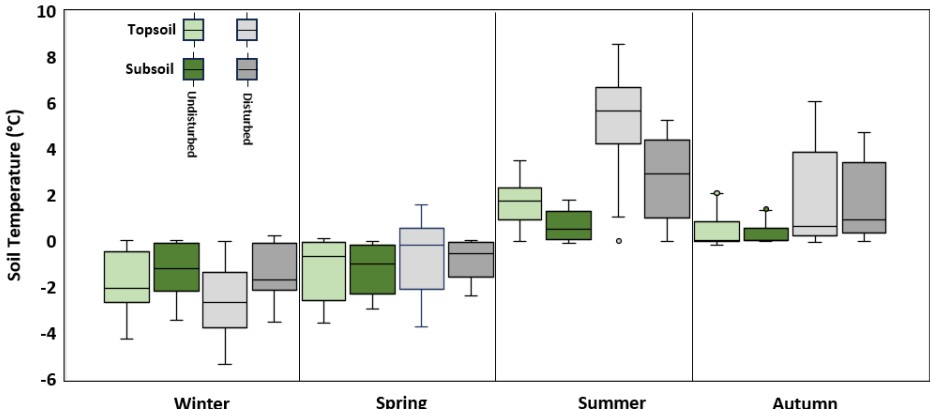


**Figure 2. Seasonal soil temperature patterns. The seasons were delineated as winter (Nov–Mar),**
**spring (Apr and May), summer (Jun–Aug), and autumn (Sep and Oct). Soil temperature are**
**average daily values from the topsoil and subsoil layers at the 8 chamber plots: 4 at the undisturbed**
**site and 4 at the disturbed site. The range of the boxplot represents the first and third quartiles,**
**while the central line signifies the median. The whiskers of the box extend to the minimum and**
**maximum values, with outliers represented by circles.**

VWC values ranged from 0.29+-0.00 m³/m³ to of 0.47+-0.00 m³/m³ (1 Jun to 31 Oct 2024) (SI
Figure S1). Subsoil layer exhibited elevated mean seasonal VWC values, as compared to the topsoil





layer, at both locations ($p$ values from 0.04 to < 0.001, ANOVA). Average maximum seasonal thaw depth
exhibited significant differences between the two locations ($p$ values 0.02, ANOVA), ranging from $58 \pm 3$
cm or $143 \pm 29$ cm at the undisturbed site or disturbed site, respectively (Figure 3). While the maximum
seasonal thaw depth remained relatively consistent across the undisturbed plots, ranging from 50 cm to 61
cm, the disturbed plot displayed a larger range in maximum thaw depth, varying from 82 cm to 204 cm.

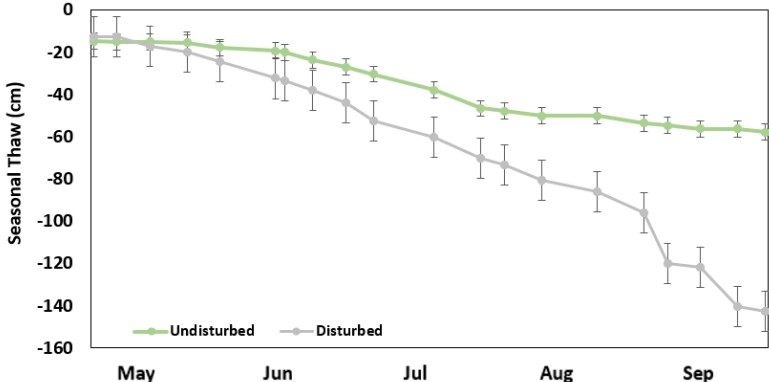


**Figure 3. Average seasonal thaw depth at the undisturbed and disturbed sites measured using**
**manual frost probe measurements from 12 May to 3 Oct 2023. Error bars are standard error**.

**3.2 Soil properties**

The disturbance treatment had a significant effect on LOI (a proxy for SOM content) in the

subsoil layer, but not the topsoil layer (SI Figure S2). There was no significant effect of sampling date or
significant interaction of sampling date and disturbance on LOI. Mean total C content for subsoil in the
disturbed treatment was 2.7% (averaged across LiCor chamber plots and sampling dates), significantly
greater than that of the undisturbed treatment mean of 1.7% ($p < 0.01$; ANOVA). Mean LOI for the
topsoil layer was 0.122 and 0.142 for disturbed and undisturbed, respectively (not significant).

Similarly, the disturbance treatment was a significant factor for soil total C content and total N

content in the subsoil layer ($p < 0.01$ and $p < 0.05$, respectively; ANOVA), but not the topsoil layer
(Figure 4), and there was no significant effect of sampling date or significant interaction of sampling date
and disturbance on either C or N. Mean total C content for subsoil in the disturbed treatment was 2.7%
(averaged across LiCor chamber plots and sampling dates), significantly greater than that of the
undisturbed treatment mean of 1.7% ($p < 0.01$; ANOVA). Mean total C content for the topsoil layer was
4.7% and 5.8% for disturbed and undisturbed, respectively (not significant). Mean total N content for



subsoil in the disturbed treatment was 0.16%, significantly greater than that of the undisturbed treatment
mean of 0.13% ($p < 0.01$; ANOVA). Mean total N content for the topsoil was 0.19% and 0.25% for
disturbed and undisturbed, respectively (not significant).

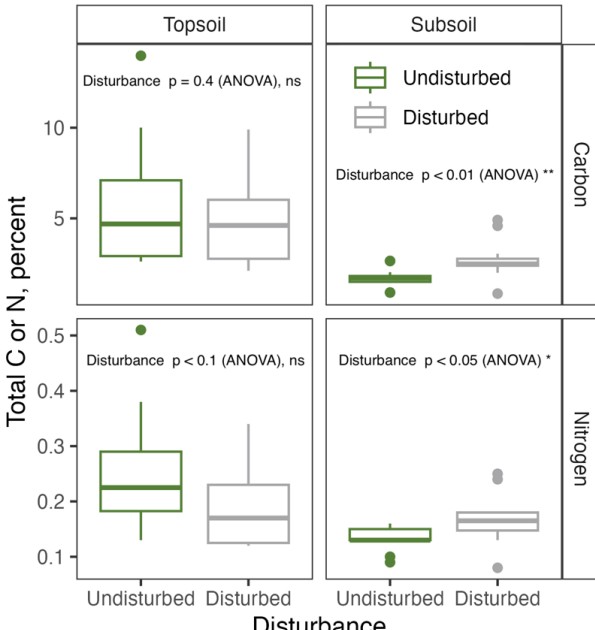


**Figure 4. Soil total carbon and total nitrogen, by disturbance treatment. Boxplots represent all**
**three sampling dates and four LiCor chamber plots, within each disturbance treatment**.


Similar to the response of LOI, total C, and total N, soil pH was significantly different in the
subsoil layer, depending on disturbance treatment; the effect of disturbance on the topsoil layer was not
significant (SI Figure S3). There was no significant effect of sampling date or significant interaction of
sampling date and disturbance on soil pH. Mean pH for subsoil in the disturbed treatment was 4.63
(averaged across chambers and sampling dates), significantly greater than that of the undisturbed
treatment mean of 4.11 ($p < 0.01$; ANOVA). Mean pH for the topsoil layer was 3.78 and 3.79 for
disturbed and undisturbed, respectively (not significant).
**3.3 Soil microbial community composition and diversity**
We analyzed the effect of disturbance on microbial community composition and diversity to
better understand the microbial role in soil respiration. The dominant bacterial phyla included



*Proteobacteria*, *Acidobacteria*, *Actinobacteria*, *Verrucomicrobia*, and *Chloroflexi*, which together
comprised over 75% of relative abundance (SI Figure S4). For fungi, 54% of relative abundance was
comprised of *Ascomycota* (mostly classes *Leotiomycetes* and *Archaeorhizomycetes*), and 40% was
*Basidiomycota* (mostly class *Agaricomycetes*) (SI Figure S5). Differential abundance testing only
identified several dozen taxa (after filtering for somewhat higher abundance taxa) that were either
significantly enriched or depleted under disturbance (see SI Figures S6 and S7). Notably, several
Chloroflexi and Dormibacterota (a newly named phylum, previously identified as Chloroflexi) ASV's are
depleted under disturbance relative to the undisturbed condition in subsoil, particularly at the February
sampling date (SI Figure S6).
Mean estimated bacterial richness across the dataset was 711.5 ASVs, with a significant effect of
disturbance (SI Figure S8). Overall, the disturbed samples had 30% higher richness compared to the
undisturbed samples ($p < 0.01$, ANOVA); within each date and soil layer combination, only the February
2023 subsoil samples demonstrated a significant effect of disturbance ($p < 0.01$, ANOVA). There was no
effect of sampling date or soil layer, or interaction amongst the factors. Mean estimated fungal richness
across the dataset was 165 ASVs; there was no effect of disturbance, sampling date, soil layer, or
interaction amongst the factors (SI Figure S8).
Using Bray-Curtis dissimilarities to evaluate beta diversity (Figure 5), there was a significant
effect of disturbance regime on both bacterial and fungal communities in both topsoil and subsoil layers
($p < 0.001$ for all; $R^2 = 0.158, 0.178, 0.186, 0.115$ for bacterial topsoil, bacterial subsoil, fungal topsoil,
and fungal subsoil communities, respectively; PERMANOVA). Sampling date did not have a significant
effect.



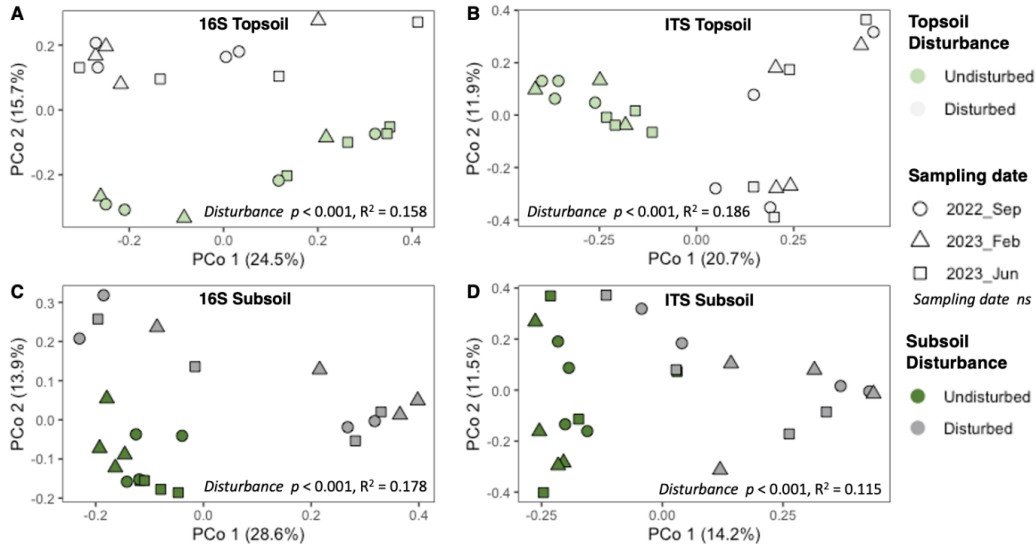

**Figure 5. Principal coordinates analysis of Bray-Curtis dissimilarities of relative abundance data following rarefaction. Panels (A) and (C) represent the bacterial community (16S marker), and panels (B) and (D) represent the fungal community composition (ITS marker). The top panels (A & B; lighter shades) represent the topsoil layer community composition, while the bottom panels (C & D; darker shades) represent the subsoil community composition.**

### 3.4 Soil Respiration

The total soil respiration showed strong seasonal and disturbance level patterns. The undisturbed site had an average annual soil efflux of $1.81 \pm 0.09$ g C-CO$_2$ m$^{-2}$ d$^{-1}$, while the disturbed site had an average annual soil efflux of $2.07 \pm 0.11$ g C-CO$_2$ m$^{-2}$ d$^{-1}$. On July 20, the undisturbed site recorded the highest average daily flux of 9.27 g C-CO$_2$ m$^{-2}$ d$^{-1}$. The peak seasonal soil efflux occurred during the summer season (SI Figure S9), with seasonal mean daily fluxes reaching $4.01 \pm 0.12$ g C-CO$_2$ m$^{-2}$ d$^{-1}$ at the undisturbed and $4.81 \pm 0.17$ g C-CO$_2$ m$^{-2}$ d$^{-1}$ at the disturbed site ($p < 0.0001$, ANOVA; Figure 6). This was followed by autumn, where seasonal mean values were lower at $1.79 \pm 0.13$ g C-CO$_2$ m$^{-2}$ d$^{-1}$ at the undisturbed and $1.91 \pm 0.15$ g C-CO$_2$ m$^{-2}$ d$^{-1}$ at the disturbed (not statistically different). Spring showed a decrease with $1.46 \pm 0.14$ g C-CO$_2$ m$^{-2}$ d$^{-1}$ at the undisturbed and $1.41 \pm 0.13$ g C-CO$_2$ m$^{-2}$ d$^{-1}$ at the disturbed site (not statistically different), while the lowest efflux was recorded during winter with $0.33 \pm 0.01$ g C-CO$_2$ m$^{-2}$ d$^{-1}$ at the undisturbed and $0.39 \pm 0.13$ g C-CO$_2$ m$^{-2}$ d$^{-1}$ at the disturbed site ($p < 0.01$, ANOVA). CO$_2$ flux was significantly different between the undisturbed and disturbed sites in the winter ($p = 0.01$ ANOVA) and summer ($p < 0.01$, ANOVA) and revealed no statistically significant efflux differences between the two sites in the shoulder seasons.



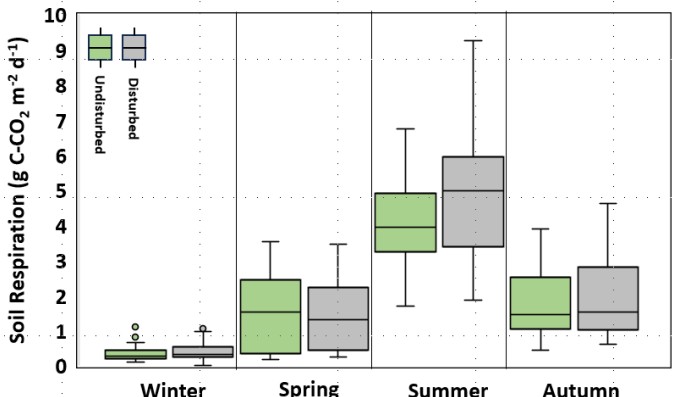

**Figure 6. Seasonal soil respiration patterns observed at the undisturbed and disturbed sites. Soil respiration emissions are average daily fluxes from the 8 long-term chambers: 4 at undisturbed site and 4 at disturbed site. The range of the boxplot represents the first and third quartiles, while the central line signifies the median. The whiskers of the box extend to the minimum and maximum values, with outliers represented by circles.**


**3.5 Linear regression analysis to determine important variables contributing to soil efflux**
A linear regression analysis was performed on the mean daily averages to investigate the seasonal
correlation between soil efflux and various factors, including air temperature, barometric pressure,
seasonal thaw depth, topsoil and subsoil temperature, and volumetric water content (VWC) at both
undisturbed and disturbed sites. The presence of disturbance did not have a significant effect ($p$ values 0.3
to 0.9, ANOVA) on the relationship between soil efflux and the aforementioned variables between the two
sites. The analysis revealed that overall soil and air temperature exhibited the strongest correlation with
soil respiration across all sites during different seasons. Soil and air temperatures showed the strongest
correlations with soil efflux across all seasons. Seasonal $R^2$ values for soil temperatures ranged from
0.21–0.65 (winter), 0.87–0.92 (spring), 0.52–0.79 (summer), and 0.85–0.93 (autumn). Air temperature
also correlated strongly with soil efflux in spring ($R^2 = 0.74$–0.77) and autumn ($R^2 = 0.83$–0.87).
Moderate correlations were observed between soil efflux and thaw depth at both sites, with $R^2$ values
ranging from 0.39–0.50 in spring, 0.41–0.43 in summer, and 0.74 in autumn.
The correlation between soil efflux and topsoil and subsoil VWC varied during the summer and
fall seasons, as well as across the two soil layers. At the undisturbed site, in summer, moderate correlation
was found in the topsoil ($R^2 = 0.38$) and strong in the subsoil ($R^2 = 0.69$). In autumn, correlations were
strong in the topsoil ($R^2 = 0.62$) but weaker in the subsoil ($R^2 = 0.22$). At the disturbed site, in summer, the
correlation was weak in the topsoil ($R^2 = 0.09$) but strong in the subsoil ($R^2 = 0.64$). In autumn, weak



correlations were observed in both layers ($R^2$ = 0.28 in topsoil and 0.10 in subsoil). Regression analysis
showed weak to no correlation between soil efflux and barometric pressure during winter and autumn ($R^2$
= 0.02–0.12) and no correlation in spring and summer. A slightly higher correlation was noted in winter
when using hourly averages ($R^2$ = 0.23 undisturbed, 0.18 disturbed).
**3.6 Random forest efflux modeling**

The RF model accurately described the effects of disturbance with the variation in soil respiration
at both locations, showing strong confidence levels with high $R^2$ and moderate to low mean absolute error
(MAE) values. Specifically, the model's $R^2$ values and corresponding MAE were 0.95 (0.14 MAE) for
winter, 0.80 (0.38 MAE) for spring, 0.94 (0.16 MAE) for summer, and 0.82 (0.04 MAE) for autumn at the
undisturbed site, and 0.95 (0.12 MAE) for winter, 0.83 (0.52 MAE) for spring, 0.96 (0.13 MAE) for
summer, and 0.84 (0.05 MAE) for autumn at the disturbed site.

The predictors' relative importance (RI) varied across seasons and sites (Figure 7). In the winter
model at the undisturbed site, air temperature emerged as the most influential predictor, accounting for
53.6% RI. Conversely, at the disturbed site, the subsoil temperature exhibited the highest predictive power
with 49.6% RI (figure 7). For the spring model, soil temperature was the best predictor at both sites, the
topsoil layer temperature exhibited a slightly greater predictive power at the undisturbed site, whereas the
subsoil layer temperature proved to be more influential at the disturbed site. In the undisturbed site's
summer model, the soil moisture and temperature as well as the seasonal thaw depth variables exhibit
similar RI values, ranging from 15.7 to 20.2%. Among these variables, air temperature has the lowest
predictive power, with an RI of 10.3%. Conversely, at the disturbed site, the topsoil layer temperature
stands out as the strongest predictor, with an RI of 44.6%. It is followed by the topsoil VWC, which has
an RI of 13.7%. The undisturbed site's autumn model indicates that subsoil layer VWC has the highest
predictive power at 24.3% RI, followed by subsoil layer temperature at 18.9% RI, and topsoil layer
temperature at 18.7% RI. However, the disturbed autumn model shows that subsoil layer temperature is
the most influential factor in predicting $CO_2$ efflux with 24.3% RI and topsoil layer (21.1% RI) and air
(18.9% RI) also playing significant roles.



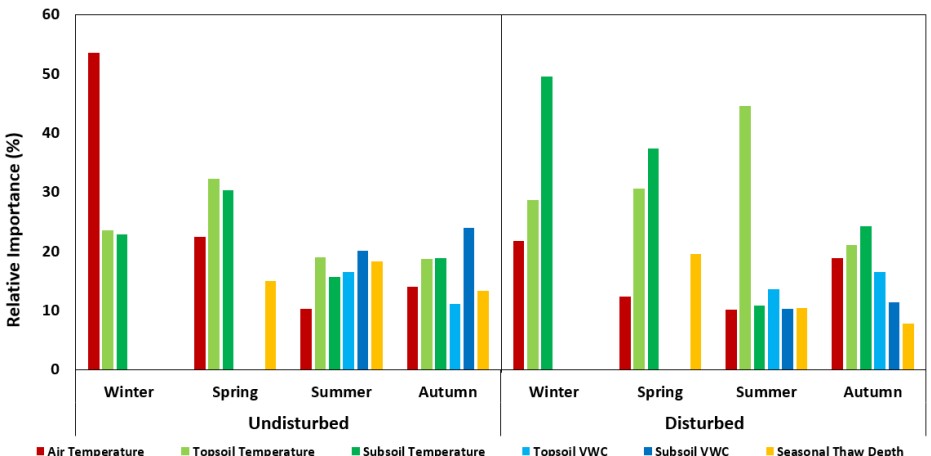

**Figure 7. Relative importance (%) of variables to predict CO₂ efflux. Missing or static input variables were not measured for importance.**

Throughout the different seasons, various factors contribute to the varying degrees of relative importance regarding soil respiration. These factors include air temperature, the temperature of both topsoil and subsoil layers, and volumetric water content, which pertains only to subsoil. This information is presented in Table 1.

**Table 1. Tabulation of the most important predictor variables to explain variability in the soil efflux data between disturbed and undisturbed sites as a function of season.**

| Season | Plot | Highest Predictor | RI |
|--------|------|-------------------|-----|
| Winter | Undisturbed | Air temperature | 53.6 |
| Winter | Disturbed | Soil layer temperature (subsoil) | 49.6 |
| Spring | Undisturbed | Soil layer temperature (topsoil) | 32.2 |
| Spring | Disturbed | Soil layer temperature (subsoil) | 37.4 |
| Summer | Undisturbed | Volumetric water content (topsoil) | 20.2 |
| Summer | Disturbed | Soil layer temperature (topsoil) | 44.6 |
| Autumn | Undisturbed | Volumetric water content (subsoil) | 24.0 |
| Autumn | Disturbed | Soil layer temperature (subsoil) | 24.3 |

## 4. Discussion





Our findings show that the temperatures of the active layer soil are markedly elevated at the
disturbed site in comparison to the undisturbed site. Previous research has indicated that this increase in
temperature is primarily due to a decrease in insulating organic matter (Gordon *et al*., 1987; Lytle and
Cronan, 1998; Amiro, 2001; Halim *et al*., 2024) and alterations in surface albedo resulting from
disturbance (Amiro, 2001). The consequences of these modified thermal conditions are significant, as
heightened soil temperatures may expedite permafrost degradation and affect biogeochemical cycles
(Chatterjee *et al*., 2008) contributing to a positive feedback cycle. It stands to reason that winter was the
only season during which soil temperatures were elevated regardless of organic content at the undisturbed
site (Figure 2), as the unaltered surface vegetation provided a layer of insulation against the low air
temperature. Furthermore, disturbance notably influenced total C, total N (Figure 4), and soil pH within
the subsoil layer.
The effects of disturbance on microbial communities were substantial, with significant alterations
noted in both bacterial and fungal compositions. Notably, the date of sampling did not exert a significant
impact on community structure, indicating that the disturbance itself serves as the principal factor driving
these changes. This finding is consistent with earlier studies that suggest soil disturbances can disrupt
microbial habitats, resulting in a reorganization of community dynamics (Chatterjee *et al*., 2008). Such
changes can lead to cascading effects on ecosystem functions, including nutrient cycling and the
decomposition of organic matter. We found that the disturbance-driven changes, led to higher average soil
temperatures and increased concentrations of mean SOM (LOI) (by a factor of 1.8) (SI Figure S2), mean
total carbon (by a factor of 1.6) (Figure 4), and mean total nitrogen (by a factor of 1.2) (Figure 4) in the
subsoil which positively influenced the mean annual soil respiration rates, resulting in a 14.4% increase
when compared to the undisturbed site.
The variability in soil efflux data between disturbed and undisturbed sites as a function of season
reveals distinct patterns influenced by key abiotic factors. In winter, air temperature emerged as the most
significant predictor for the undisturbed sites, likely due to its direct impact on microbial activity and
respiration rates. In contrast, the disturbed site during the same season showed a stronger correlation with
the subsoil layer temperature, suggesting that disturbances may alter the thermal dynamics of the soil
profile, making soil temperature a more crucial determinant. During spring, topsoil layer temperature was
the primary predictor for the undisturbed site, highlighting the importance of organic matter
decomposition driven by temperature changes. For the disturbed site, the subsoil layer temperature
remained the dominant factor, indicating that disturbances may have disrupted the organic layer, shifting
the focus to the subsoil layer's thermal conditions.



In the summer months, the volumetric water content in the subsoil layer emerged as the principal
predictor for the undisturbed site, highlighting the essential role of moisture availability in influencing
microbial activity and respiration during this warmer period. Conversely, the disturbed site exhibited a
more pronounced correlation with the temperature of the topsoil layer, indicating that disturbances may
have modified the soil's hydrological characteristics by increasing thaw depth and enhancing drainage,
which resulted in a reduced volumetric water content and rendered temperature a more significant factor
than moisture. As autumn approached, the volumetric water content in the subsoil layer reestablished
itself as the key predictor for the undisturbed site, reaffirming the ongoing significance of moisture for
microbial functions as temperatures began to decline. In contrast, for the disturbed site, the temperature of
the subsoil layer continued to be the prevailing influence, underscoring the enduring effects of
disturbances on the thermal dynamics of the soil.
Overall, the observed patterns indicate that soil disturbances fundamentally alter the relationships
between abiotic factors and $CO_2$ efflux. By reshaping thermal profiles and hydrological properties,
disturbances intensify microbial activity across seasons, thereby increasing $CO_2$ emissions.

**Conclusion**

As warming continues, thawing permafrost and degrading pristine areas in arctic and sub-arctic
regimes, there will be a higher degree of ecological response that occurs, particularly related to soil
respiration. Areas that, in theory, are undisturbed will begin to mirror anthropogenically disturbed areas in
terms of greenhouse gas efflux. This study provided valuable insights into the effects of disturbance on
critical soil properties and their influence on soil respiration. Our research indicates notable changes in
soil temperature, volumetric water content, the composition of bacterial and fungal communities, total
mean SOM, C and N levels and pH values in the subsoil layer, and the depth of the active layer as a result
of disturbances, which subsequently led to higher soil respiration rates. Soil respiration was primarily
regulated by temperature, (air and soil) while factors such as soil volumetric water content and the depth
of the active layer also contributed, with their relative importance varying throughout the different
seasons. These findings underscore the complex interplay between seasonal variations, soil disturbance,
and abiotic factors in determining soil respiration rates. Understanding these relationships is essential for
accurate modeling of carbon cycling and for developing effective strategies to mitigate the impacts of soil
disturbances on ecosystem functions. Both natural and anthropogenic disturbances can lead to a marked
rise in the emission of carbon dioxide and other greenhouse gases into the atmosphere. Neglecting to
account for these disturbances may result in a considerable underestimation of the role of soils in global



carbon cycles. Future investigations should concentrate on the long-term consequences of these dynamics,
especially considering ongoing warming change and its impact on permafrost regions.
*Data availability.* The datasets produced and/or examined in the present study can be obtained from the
corresponding author upon reasonable request.
*Author Contributions.* DAV conceptualized the study and managed the data collection, RAB provided
funding, DAV, AJB, and WBB collected the soil samples, DAV and DB performed the soil respiration
analysis, JRW performed the soil property and microbial community analyses. DAV drafted the initial
manuscript and all authors contributed to revisions.
*Competing Interests.* The authors have no relevant financial or non-financial interests to disclose.
*Acknowledgements.* The authors express their gratitude to Ms. Elizabeth Corriveau for her work in
conducting quality assurance and quality control on the soil respiration time series data and to Ms. Anne
Katula for her efforts in processing the soil samples for analyses related to microbial composition and soil
properties.
*Financial support.* This work was founded by PE 0602144A Program Increase 'Defense Resiliency
Platform Against Extreme Cold Weather'.

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

Glöckner, F.O., 2014. All-species Living Tree Project (LTP)" taxonomic frameworks. *Nucleic Acids
Res*, *42*, pp.D643-D648.