# Peer review of "Effects of permafrost thaw on seasonal soil CO2 efflux dynamics in a boreal"

_EGUsphere, 2025_

## Editor Comment (EC1)

**Major comments for Authors:**

There is often mention of microbial activity in the manuscript, but the authors are measuring both autotrophic and heterotrophic respiration. How does this link to microbial activity? How does it link to the measures of microbial community structure? There seem to be a lot of measurements that don't necessarily fit together.

Disturbance – needs to be a much more thorough discussion of the type of disturbance. How does it influence e.g. the location/distribution of the organic layer? How might this influence your measurements of carbon and organic matter distribution? Need to know more than it was "thawed permafrost." Is the permafrost continuous or discontinuous? Please rationalize the different sampling depths in disturbed vs. undisturbed.

How relevant is all the peatland literature that is cited? Are all these forests dominated by peat? Peatlands have issues with e.g. water table depth/movement that influence gas fluxes. Are those processes important here?

The authors make a statement that their work "reveals the fundamental mechanisms of soil respiration." I don't see this. Maybe "fundamental controls"??

For discussion of carbon and SOM distribution, be careful about use of words like concentration and content. Content suggests amounts per unit area. To do this, you need bulk density data. I don't think you have those data based on what I see in the manuscript.

How did you calculate microbial diversity? I saw data for richness. I see – Bray/curtiss dissimilarities. Is that your estimate of diversity? Can you describe the diversity results directly (ie this was more diverse than that)?

Random Forest shows an impact of disturbance on soil CO2, but the regression analysis does not. So which do we believe?? Also, the use of respiration data to model disturbance (section 3.6) was not very clear and a bit hard to follow. You also state early in the Discussion that disturbance is key, yet your analyses do not seem to show much support for this.

There is no connection of your results to previous literature. There needs to be a bit more discussion of how your results fit into the pre-existing literature and our understanding of the controls on these fluxes.